# TRIGGER HUNTING WITH A TOPOLOGICAL PRIOR FOR TROJAN DETECTION

**Xiaoling Hu** *
Stony Brook University

**Xiao Lin, Michael Cogswell, Yi Yao & Susmit Jha**
SRI International

**Chao Chen**
Stony Brook University

## ABSTRACT

Despite their success and popularity, deep neural networks (DNNs) are vulnerable when facing backdoor attacks. This impedes their wider adoption, especially in mission critical applications. This paper tackles the problem of Trojan detection, namely, identifying Trojaned models – models trained with poisoned data. One popular approach is reverse engineering, i.e., recovering the triggers on a clean image by manipulating the model's prediction. One major challenge of reverse engineering approach is the enormous search space of triggers. To this end, we propose innovative priors such as **diversity** and **topological simplicity** to not only increase the chances of finding the appropriate triggers but also improve the quality of the found triggers. Moreover, by encouraging a diverse set of trigger candidates, our method can perform effectively in cases with unknown target labels. We demonstrate that these priors can significantly improve the quality of the recovered triggers, resulting in substantially improved Trojan detection accuracy as validated on both synthetic and publicly available TrojAI benchmarks.

## 1 INTRODUCTION

Deep learning has achieved superior performance in various computer vision tasks, such as image classification (Krizhevsky et al., 2012), image segmentation (Long et al., 2015), object detection (Girshick et al., 2014), etc. However, the vulnerability of DNNs against backdoor attacks raises serious concerns. In this paper, we address the problem of *Trojan attacks*, where during training, an attacker injects *polluted samples*. While resembling normal samples, these polluted samples contain a specific type of perturbation (called triggers). These polluted samples are assigned with *target labels*, which are usually different from the expected class labels. Training with this polluted dataset results in a *Trojaned model*. At the inference stage, a Trojaned model behaves normally given clean samples. But when the trigger is present, it makes unexpected, yet consistently incorrect predictions.

One major constraint for Trojan detection is the limited access to the polluted training data. In practice, the end-users, who need to detect the Trojaned models, often only have access to the weights and architectures of the trained DNNs. State-of-the-art (SOTA) Trojan detection methods generally adopt a *reverse engineering* approach (Guo et al., 2019; Wang et al., 2019; Huster & Ekwedike, 2021; Wang et al., 2020b; Chen et al., 2019c; Liu et al., 2019). They start with a few clean samples, using either gradient descent or careful stimuli crafting, to find a potential trigger that alters model prediction. Characteristics of the recovered triggers along with the associated network activations are used as features to determine whether a model is Trojaned or not.

Trojan triggers can be of arbitrary patterns (e.g., shape, color, texture) at arbitrary locations of an input image (e.g., Fig. 1). As a result, one major challenge of the reverse engineering-based approach is the enormous search space for potential triggers. Meanwhile, just like the trigger is unknown, the target label (i.e., the class label to which a triggered model predicts) is also unknown in practice. Gradient descent may flip a model's prediction to the closest alternative label, which may not necessarily be the target label. This makes it even more challenging to recover the true trigger. Note that many existing methods (Guo et al., 2019; Wang et al., 2019; 2020b) require a target label. These methods achieve

---

*Email: Xiaoling Hu (xiaolhu@cs.stonybrook.edu).

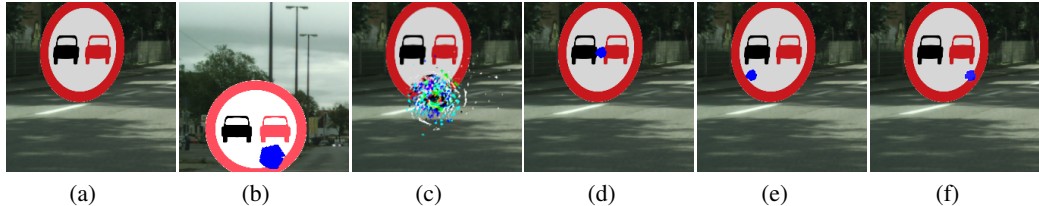

| (a) | (b) | (c) | (d) | (e) | (f) |

Figure 1: Illustration of recovered triggers: **(a)** clean image, **(b)** poisoned image, **(c)** image with a trigger recovered without topological prior, **(d)-(f)** images with candidate triggers recovered with the proposed method. Topological prior contributes to improved compactness. We run the trigger reconstruction for multiple rounds with a diversity prior to ensure a diverse set of trigger candidates.

target label independence by enumerating through all possible labels, which can be computationally prohibitive especially when the label space is huge.

We propose a novel target-label-agnostic reverse engineering method. First, to improve the quality of the recovered triggers, we need a prior that can localize the triggers, but in a flexible manner. We, therefore, propose to enforce a *topological prior* to the optimization process of reverse engineering, i.e., the recovered trigger should have fewer connected components. This prior is implemented through a topological loss based on the theory of persistent homology (Edelsbrunner & Harer, 2010). It allows the recovered trigger to have arbitrary shape and size. Meanwhile, it ensures the trigger is not scattered and is reasonably localized. See Fig. 1 for an example – comparing (d)-(f) vs. (c).

As a second contribution, we propose to reverse engineer *multiple diverse trigger candidates*. Instead of running gradient decent once, we run it for multiple rounds, each time producing one trigger candidate, e.g., Fig. 1 (d)-(f). Furthermore, we propose a *trigger diversity loss* to ensure the trigger candidates to be sufficiently different from each other (see Fig. 2). Generating multiple diverse trigger candidates can increase the chance of finding the true trigger. It also mitigates the risk of unknown target labels. In the example of Fig. 2, the first trigger candidate flips the model prediction to a label different from the target, while only the third candidate hits the true target label.

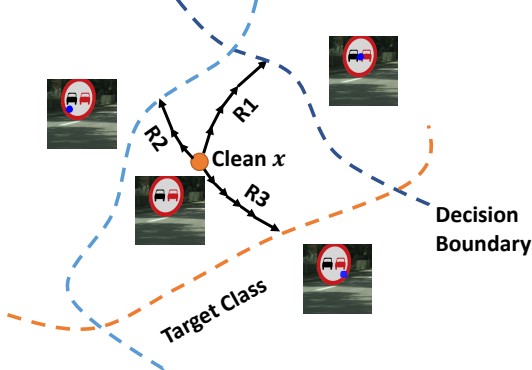

Figure 2: Illustration of generating a diverse set of trigger candidates to increase the chance of finding the true trigger, especially for scenarios with unknown target labels.

Generating multiple trigger candidates, however, adds difficulties in filtering out already-subtle cues for Trojan detection. We also note reverse engineering approaches often suffer from false positive triggers such as adversarial perturbations or direct modification of the crucial objects of the image[1]. In practice, we systematically extract a rich set of features to describe the characteristics of the reconstructed trigger candidates based on geometry, color, and topology, as well as network activations. A Trojan-detection network is then trained to detect Trojaned models based on these features. Our main contributions are summarized as follows:

- We propose a topological prior to regularize the optimization process of reverse engineering. The prior ensures the locality of the recovered triggers, while being sufficiently flexible regarding the appearance. It significantly improves the quality of the reconstructed triggers.

- We propose a diversity loss to generate multiple diverse trigger candidates. This increases the chance of recovering the true trigger, especially for cases with unknown target labels.

- Combining the topological prior and diversity loss, we propose a novel Trojan detection framework. On both synthetic and public TrojAI benchmarks, our method demonstrates substantial improvement in both trigger recovery and Trojan detection.

---

[1]Modification of crucial objects is usually not a valid trigger strategy; it is too obvious to end-users and is against the principle of adversaries.

## 2   RELATED WORK

**Trojan detection.** Many Trojan detection methods have been proposed recently. Some focus on detecting poisoned inputs via anomaly detection (Chou et al., 2020; Gao et al., 2019; Liu et al., 2017; Ma & Liu, 2019). For example, SentiNet (Chou et al., 2020) tries to identify adversarial inputs, and uses the behaviors of these adversarial inputs to detect Trojaned models. Others focus on analyzing the behaviors of the trained models (Chen et al., 2019a; Guo et al., 2019; Shen et al., 2021; Sun et al., 2020). Specifically, Chen et al. (2019a) propose the Activation Clustering (AC) methodology to analyze the activations of neural networks to determine if a model has been poisoned or not.

While early works require all training data to detect Trojans (Chen et al., 2019a; Gao et al., 2019; Tran et al., 2018), recent approaches have been focusing on a more realistic setting – when one has limited access to the training data. A particular promising direction is reverse engineering approaches, which recover Trojan triggers with only a few clean samples. Neural cleanse (NC) (Wang et al., 2019) develops a Trojan detection method by identifying if there is a trigger that would produce misclassified results when added to an input. However, as pointed out by (Guo et al., 2019), NC becomes futile when triggers vary in terms of size, shape, and location.

Since NC, different approaches have been proposed, extending the reverse engineering idea. Using a conditional generative model, DeepInspect (Chen et al., 2019c) learns the probability distribution of potential triggers from a model of interest. Kolouri et al. (2020) propose to learn universal patterns that change predictions of the model (called Universal Litmus Patterns (ULPs)). The method is efficient as it only involves forward passes through a CNN and avoids backpropagation. ABS (Liu et al., 2019) analyzes inner neuron behaviors by measuring how extra stimulation can change the network's prediction. Wang et al. (2020b) propose a data-limited TrojanNet detector (TND) by comparing the impact of per-sample attack and universal attack. Guo et al. (2019) cast Trojan detection as a non-convex optimization problem and it is solved through optimizing an objective function. Huster & Ekwedike (2021) solve the problem by observing that, compared with clean models, adversarial perturbations transfer from image to image more readily in poisoned models. Zheng et al. (2021) inspect neural network structure using persistent homology and identify structural cues differentiating Trojaned and clean models.

Existing methods are generally demanding on training data access, neural network architectures, types of triggers, target class, etc. This limits their deployment to real-world applications. As for reverse engineering approaches, it remains challenging, if not entirely infeasible, to recover the true triggers. We propose a novel reverse engineering approach that can recover the triggers with high quality using the novel diversity and topological prior. Our method shares the common benefit of reverse engineering methods; it only needs a few clean input images per model. Meanwhile, our approach is agnostic of model architectures, trigger types, and target labels.

**Topological data analysis and persistent homology.** Topological data analysis (TDA) is a field in which one analyzes datasets using topological tools such as persistent homology (Edelsbrunner & Harer, 2010; Edelsbrunner et al., 2000). The theory has been applied to different applications (Wu et al., 2017; Kwitt et al., 2015; Wong et al., 2016; Chazal et al., 2013; Ni et al., 2017; Adams et al., 2017; Bubenik, 2015; Varshney & Ramamurthy, 2015; Hu et al., 2019; 2021; Zhao et al., 2020; Yan et al., 2021; Wu et al., 2020).

As the advent of deep learning, some works have tried to incorporate the topological information into deep neural networks, and the differentiable property of persistent homology make it possible. The main idea is that the persistence diagram/barcodes can capture all the topological changes, and it is differentiable to the original data. Hu et al. (2019) first propose a topological loss to learn to segment images with correct topology, by matching persistence diagrams in a supervised manner. Similarly, Clough et al. (2020) use the persistence barcodes to enforce a given topological prior of the target object. These methods achieve better results especially in structural accuracy. Persistent-homology-based losses have been applied to other imaging (Abousamra et al., 2021; Wang et al., 2020a) and learning problems (Hofer et al., 2019; 2020; Carrière et al., 2020; Chen et al., 2019b).

The aforementioned methods use topological priors in supervised learning tasks (namely, segmentation). Instead, in this work, we propose to leverage the topological prior in an unsupervised setting; we use a topological prior for the reverse engineering pipeline to reduce the search space of triggers and enforce the recovered triggers to have fewer connected components.

## 3 METHOD

Our reverse engineering framework is illustrated in Fig. 3. Given a trained DNN model, either clean or Trojaned, and a few clean images, we use gradient descent to reconstruct triggers that can flip the model's prediction. To increase the quality of reconstructed triggers, we introduce novel diversity loss and topological prior. They help recover multiple diverse triggers of high quality.

The common hypothesis of reverse engineering approaches is that the reconstructed triggers will appear different for Trojaned and clean models. To fully exploit the discriminative power of the reconstructed triggers for Trojan detection, we extract features based on trigger characteristics and associated network activations. These features are used to train a classifier, called the Trojan-detection network, to classify a given model as Trojaned or clean.

We note the discriminative power of the extracted trigger features, and thus the Trojan-detection network, are highly dependent on the quality of the reconstructed triggers. Empirical results will show the proposed diversity loss and topological prior are crucial in reconstructing high quality triggers, and ensures a high quality Trojan-detection network. We will show that our method can learn to detect Trojaned models even when trained with a small amount of labeled DNN models.

For the rest of this section, we mainly focus on the reverse engineering module. We also add details of the Trigger feature extraction to Sec. 3.3 and details of the Trojan-detection network to Sec. A.1.

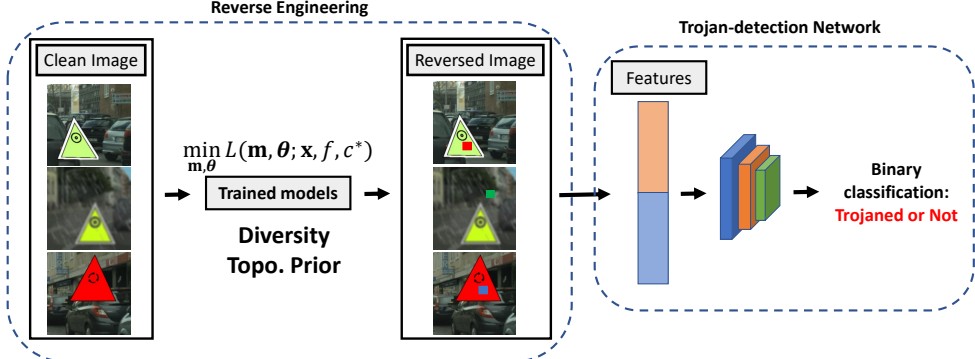

Figure 3: Our Trojan detection framework.

### 3.1 REVERSE ENGINEERING OF MULTIPLE DIVERSE TRIGGER CANDIDATES

Our method is based on the existing reverse engineering pipeline first proposed by Neural Cleanse (Wang et al., 2019). Given a trained DNN model, let $f(\cdot)$ be the mapping from an input clean image $\mathbf{x} \in \mathbb{R}^{3 \times M \times N}$ to the output $\mathbf{y} \in \mathbb{R}^K$ with $K$ classes, where $M$ and $N$ denote the height and width of the image, respectively. Denote by $f_k(\cdot)$ the $k$-th output of $f$. The predicted label $c^*$ is given by $c^* = \arg\max_k f_k(\mathbf{x})$, $1 \le k \le K$. We introduce parameters $\boldsymbol{\theta}$ and $\mathbf{m}$ to convert $\mathbf{x}$ into an altered sample

$$\phi(\mathbf{x}, \mathbf{m}, \boldsymbol{\theta}) = (\mathbf{1} - \mathbf{m}) \odot \mathbf{x} + \mathbf{m} \odot \boldsymbol{\theta}, \quad (1)$$

where the binary mask $\mathbf{m} \in \{0, 1\}^{M \times N}$ and the pattern $\boldsymbol{\theta} \in \mathbb{R}^{M \times N}$ determine the trigger. $\mathbf{1}$ denotes an all-one matrix. The symbol "$\odot$" denotes Hadamard product. See Fig. 4 for an illustration. We intend to find a triggered image $\hat{\mathbf{x}} = \phi(\mathbf{x}, \hat{\mathbf{m}}, \hat{\boldsymbol{\theta}})$ so that the model prediction $\hat{c} = \arg\max_k f_k(\hat{\mathbf{x}})$ is different from the prediction on the original image $c^*$.

Figure 4: $\mathbf{m}$ and $\boldsymbol{\theta}$ convert an input image $\mathbf{x}$ into an altered one $\phi(\mathbf{x}, \mathbf{m}, \boldsymbol{\theta})$. The $\odot$ is omitted here for simplification.

We find the triggered image, $\hat{\mathbf{x}}$, by minimizing a loss over the space of $\mathbf{m}$ and $\boldsymbol{\theta}$:

$$L(\mathbf{m}, \boldsymbol{\theta}; \mathbf{x}, f, c^*) = L_{flip}(\dots) + \lambda_1 L_{div}(\dots) + \lambda_2 L_{topo}(\dots) + R(\mathbf{m}), \quad (2)$$

where $L_{flip}$, $L_{div}$, and $L_{topo}$ denote the label-flipping loss, diversity loss, and topological loss, respectively. We temporarily dropped their arguments for convenience. $\lambda_1$, $\lambda_2$ are the weights to

balance the loss terms. $R(\mathbf{m})$ is a regularization term penalizing the size and range of the mask (more details will be provided in the Sec. A.1 of Appendix).

To facilitate optimization, we relax the constraint on the mask $\mathbf{m}$ and allow it to be a continuous-valued function, ranging between 0 and 1 and defined over the image domain, $\mathbf{m} \in [0, 1]^{M \times N}$. Next, we introduce the three loss terms one-by-one.

**Label-flipping loss** $L_{flip}$: The label-flipping loss $L_{flip}$ penalizes the prediction of the model regarding the ground truth label, formally:

$$L_{flip}(\mathbf{m}, \boldsymbol{\theta}; \mathbf{x}, f, c^*) = f_{c^*}(\phi(\mathbf{x}, \mathbf{m}, \boldsymbol{\theta})). \tag{3}$$

Minimizing $L_{flip}$ means minimizing the probability that the altered image $\phi(\mathbf{x}, \mathbf{m}, \boldsymbol{\theta})$ is predicted as $c^*$. In other words, we are pushing the input image out of its initial decision region.

Note that we do not specify which label we would like to flip the prediction to. This makes the optimization easier. Existing approaches often run optimization to flip the label to a target label and enumerate through all possible target labels (Wang et al., 2019; 2020b). This can be rather expensive in computation, especially with large label space.

The downside of not specifying a target label during optimization is we will potentially miss the correct target label, i.e., the label which the Trojaned model predicts on a triggered image. To this end, we propose to reconstruct multiple candidate triggers with diversity constraints. This will increase the chance of hitting the correct target label. See Fig. 2 for an illustration.

**Diversity loss** $L_{div}$: With the label-flipping loss $L_{flip}$, we flip the label to a different one from the original clean label and recover the corresponding triggers. The new label, however, may not be the same as the true target label. Also considering the huge trigger search space, it is difficult to recover the triggers with only one attempt. Instead, we propose to search for multiple trigger candidates to increase the chance of capturing the true trigger.

We run our algorithm for $N_T$ rounds, each time reconstructing a different trigger candidate. To avoid finding similar trigger candidates, we introduce the diversity loss $L_{div}$ to encourage different trigger patterns and locations. Let $\mathbf{m}_j$ and $\boldsymbol{\theta}_j$ denote the trigger mask and pattern found in the $j$-th round. At the $i$-th round, we compare the current candidates with triggers from all previous founds in terms of $L_2$ norm. Formally:

$$L_{div}(\mathbf{m}, \boldsymbol{\theta}) = -\sum_{j=1}^{i-1} ||\mathbf{m} \odot \boldsymbol{\theta} - \mathbf{m}_j \odot \boldsymbol{\theta}_j||_2. \tag{4}$$

Minimizing $L_{div}$ ensures the eventual trigger $\mathbf{m}_i \odot \boldsymbol{\theta}_i$ to be different from triggers from previous rounds. Fig. 1(d)-(f) demonstrates the multiple candidates recovered with sufficient diversity.

## 3.2 TOPOLOGICAL PRIOR

Quality control of the trigger reconstruction remains a major challenge in reverse engineering methods, due to the huge search space of triggers. Even with the regularizor $R(\mathbf{m})$, the recovered triggers can still be scattered and unrealistic. See Fig. 1(c) for an illustration. We propose a topological prior to improve the locality of the reconstructed trigger. We introduce a topological loss enforcing that the recovered trigger mask $\mathbf{m}$ to have as few number of connected components as possible. The loss is based on the theory of persistent homology (Edelsbrunner et al., 2000; Edelsbrunner & Harer, 2010), which models the topological structures of a continuous signal in a robust manner.

**Persistent homology.** We introduce persistent homology in the context of 2D images. A more comprehensive treatment of the topic can be found in (Edelsbrunner & Harer, 2010; Dey & Wang, 2021). Recall we relaxed the mask function $\mathbf{m}$ to a continuous-valued function defined over the image domain (denoted by $\Omega$). Given any threshold $\alpha$, we can threshold the image domain with regard to $\mathbf{m}$ and obtain the *superlevel set*, $\Omega^\alpha := \{p \in \Omega | \mathbf{m}(p) \geq \alpha\}$. A superlevel set can have different topological structures, e.g., connected components and holes. If we continuously decrease the value $\alpha$, we have a continuously growing superlevel set $\Omega^\alpha$. This sequence of superlevel set is called a *filtration*. The topology of $\Omega^\alpha$ continuously changes through the filtration. New connected components are born and later die (are merged with others). New holes are born and later die (are sealed up). For each topological structure, the threshold at which it is born is called its *birth time*. The threshold at which it dies is called its *death time*. The difference between birth and death time is called the *persistence* of the topological structure.

We record the lifespan of all topological structures over the filtration and encode them via a 2D point set called *persistence diagram*, denoted by $\mathrm{Dgm}(\mathbf{m})$. Each topological structure is represented by a 2D point within the diagram, $p \in \mathrm{Dgm}(\mathbf{m})$, called a *persistent dot*. We use the birth and death times of the topological structure to define the coordinates of the corresponding persistent dot. For each dot $p \in \mathrm{Dgm}(\mathbf{m})$, we abuse the notation and call the birth/death time of its corresponding topological structure as $\mathrm{birth}(p)$ and $\mathrm{death}(p)$. Then we have $p = (\mathrm{death}(p), \mathrm{birth}(p))$. See Fig. 5 for an example function $\mathbf{m}$ (viewed as a terrain function) and its corresponding diagram. There are five dots in the diagram, corresponding to five peaks in the landscape view.

To compute persistence diagram, we use the classic algorithm (Edelsbrunner & Harer, 2010; Edelsbrunner et al., 2000) with an efficient implementation (Chen & Kerber, 2011; Wagner et al., 2012). The image is first discretized into a cubical complex consisting of vertices (pixels), edges and squares. A boundary matrix is then created to encode the adjacency relationship between these elements. The algorithm essentially carries out a matrix reduction algorithm over the boundary matrix, and the reduced matrix reads out the persistence diagram.

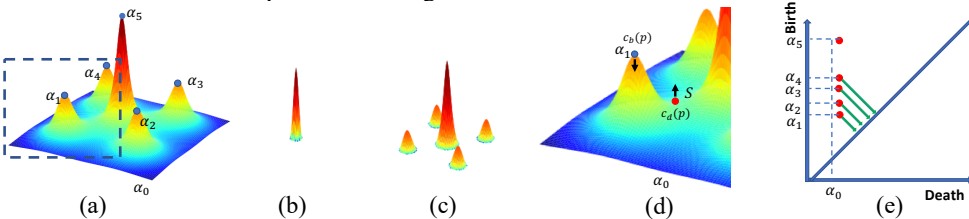

Figure 5: From the left to right: **(a)** a sample landscape for a continuous function. The values at the peaks $\alpha_0 < \alpha_1 < \alpha_2 < \alpha_3 < \alpha_4 < \alpha_5$. As we decrease the threshold, the topological structures of the superlevel set change, **(b)** and **(c)** correspond to topological structures captured by different thresholds, **(d)** highlighted region in **(a)**, **(e)** the changes are captured by the persistence diagram (right figure). We focus on the 0-dimensional topological structures (connected components). Each persistent dot in the persistence diagram denotes a specific connected component. The topological loss is introduced to reduce the connected components, which means pushing most of the persistent dots to the diagonal (along the green lines).

**Topological loss $L_{topo}$**: We formulate our topological loss based on persistent homology described above. Minimizing our loss reduces the number of connected components of triggers. We will focus on zero-dimensional topological structure, i.e., connected components. Intuitively speaking, each dot in the diagram corresponds to a connected component. The ones far away from the diagonal line are considered salient as its birth and death times are far apart. And the ones close to the diagonal line are considered trivial. In Fig. 5, there is one salient dot far away from the diagonal line. It corresponds to the highest peak. The other four dots are closer to the diagonal line and correspond to the smaller peaks. The topological loss will reduce the number of connected components by penalizing the distance of all dots from the diagonal line, except for the most salient one. Formally, the loss $L_{topo}$ is defined as:

$$L_{topo}(\mathbf{m}) = \sum_{p \in \mathrm{Dgm}(m) \setminus \{p^*\}} [\mathrm{birth}(p) - \mathrm{death}(p)]^2, \tag{5}$$

where $p^*$ denotes the persistent dot that is farthest away from the diagonal (with the highest persistence). Minimizing this loss will keep $p^*$ intact, while pushing all other dots to the diagonal line, thus making their corresponding components either disappear or merged with the main component.

**Differentiability and the gradient**: The loss function (Eq. (5)) is differentiable almost everywhere in the space of functions. To see this, we revisit the filtration, i.e., the growing superlevel set as we continuously decrease the threshold $\alpha$. The topological structures change at specific locations of the image domain. A component is born at the corresponding local maximum. It dies merging with another component at the saddle point between the two peaks. In fact, these locations correspond to critical points of the function. And the function values at these critical points correspond to the birth and death times of these topological structures. For a persistent dot, $p$, we call the critical point corresponding to its birth, $c_b(p)$, and the critical point corresponding to its death, $c_d(p)$. Then we have $\mathrm{birth}(p) = \mathbf{m}(c_b(p))$ and $\mathrm{death}(p) = \mathbf{m}(c_d(p))$. The loss function (Eq. 5) can be rewritten as a polynomial function of the function $\mathbf{m}$ at different critical points.

$$L_{topo}(\mathbf{m}) = \sum_{p \in \mathrm{Dgm}(m) \setminus \{p^*\}} [\mathbf{m}(c_b(p)) - \mathbf{m}(c_d(p))]^2. \tag{6}$$

The gradient can be computed naturally. $L_{topo}$ is a piecewise differentiable loss function over the space of all possible functions $\mathbf{m}$. In a gradient decent step, for all dots except for $p^*$, we push up the function at the death critical point $c_d(p)$ (the saddle), and push down the function value at the birth critical point $c_b(p)$ (the local maximum). This is illustrated by the arrows in Fig. 5(Middle-Right). This will kill the non-salient components and push them towards the diagonal.

### 3.3 TRIGGER FEATURE EXTRACTION AND TROJAN DETECTION NETWORK

Next we summarize the features we extract from recovered triggers. The recovered Trojan triggers can be characterized via their capability in flipping model predictions (i.e., the label-flipping loss). Moreover, they are different from adversarial noise as they tend to be more regularly shaped and are also distinct from actual objects which can be recognized by a trained model. We introduce appearance-based features to differentiate triggers from adversarial noise and actual objects, .

Specifically, for label flipping capability, we directly use the label-flipping loss $L_{flip}$ and diversity loss $L_{div}$ as features. For appearance features, we use trigger size and topological statistics as their features: 1) The number of foreground pixels divided by total number of pixels in mask $\mathbf{m}$; 2) To capture the size of the triggers in the horizontal and vertical directions, we fit a Gaussian distribution to the mask $\mathbf{m}$ and record *mean* and *std* in both directions; 3) The trigger we find may have multiple connected components. The final formulated topological descriptor includes the topological loss $L_{topo}$, the number of connected components, *mean* and *std* in terms of the size of each component.

After the features are extracted, we build a neural network for Trojan detection, which takes the bag of features of the generated triggers as inputs, and outputs a scalar score of whether the model is Trojaned or not. More details are provided in Sec. A.1 of Appendix.

## 4 EXPERIMENTS

We evaluate our method on both synthetic datasets and publicly available TrojAI benchmarks. We provide quantitative and qualitative results, followed by ablation studies, to demonstrate the efficacy of the proposed method. All clean/Trojaned models are DNNs trained for image classification.

**Synthetic datasets (Trojaned-MNIST and Trojaned-CIFAR10)**: We adopt the codes provided by NIST[2] to generate 200 DNNs (50% of them are Trojaned) trained to classify MNIST and CIFAR10 data, respectively. The Trojaned models are trained with images poisoned by square triggers. The poison rate is set as 0.2.

**TrojAI benchmarks (TrojAI-Round1, Round2, Round3 and Round4)**: These datasets are provided by US IARPA/NIST[3], who recently organized a Trojan AI competition. Polygon triggers are generated randomly with variations in shape, size, and color. Filter-based triggers are generated by randomly choosing from five distinct filters. Trojan detection is more challenging on these TrojAI datasets as compared to Triggered-MNIST due to the use of deeper DNNs and larger variations in appearances of foreground/background objects, trigger patterns etc. Round1, Round2, Round3 and Round4 have 1000, 1104, 1008 and 1008 models, respectively. Descriptions of the difference among these rounds are provided in Sec. A.2 of Appendix.

**Baselines**: We choose recently published methods including NC (Neural Cleanse) (Wang et al., 2019), ABS (Liu et al., 2019), TABOR (Guo et al., 2019), ULP (Kolouri et al., 2020), and DLTND (Wang et al., 2020b) as baselines.

**Implementation details**: We set $\lambda_1 = 1$, $\lambda_2 = 10$ and $N_T = 3$ for all our experiments (i.e., we generate 3 trigger candidates for each input image and each model). The parameters of Trojan detection network are learned using a set of clean and Trojaned models with ground truth labeling. We train

Table 1: Comparison on Trojaned-MNIST/CIFAR10.

| Method | Metric | Trojaned-MNIST | Trojaned-CIFAR10 |
|--------|--------|----------------|-------------------|
| NC | AUC | $0.57 \pm 0.07$ | $0.75 \pm 0.07$ |
| ABS | AUC | $0.63 \pm 0.04$ | $0.67 \pm 0.06$ |
| TABOR | AUC | $0.65 \pm 0.07$ | $0.71 \pm 0.05$ |
| ULP | AUC | $0.59 \pm 0.03$ | $0.55 \pm 0.03$ |
| DLTND | AUC | $0.62 \pm 0.05$ | $0.52 \pm 0.08$ |
| Ours | AUC | $\mathbf{0.88 \pm 0.04}$ | $\mathbf{0.91 \pm 0.05}$ |
| NC | ACC | $0.60 \pm 0.04$ | $0.73 \pm 0.06$ |
| ABS | ACC | $0.65 \pm 0.02$ | $0.69 \pm 0.04$ |
| TABOR | ACC | $0.62 \pm 0.04$ | $0.69 \pm 0.08$ |
| ULP | ACC | $0.57 \pm 0.02$ | $0.59 \pm 0.06$ |
| DLTND | ACC | $0.64 \pm 0.07$ | $0.55 \pm 0.07$ |
| Ours | ACC | $\mathbf{0.89 \pm 0.02}$ | $\mathbf{0.92 \pm 0.04}$ |

---

[2]https://github.com/trojai/trojai

[3]https://pages.nist.gov/trojai/docs/data.html

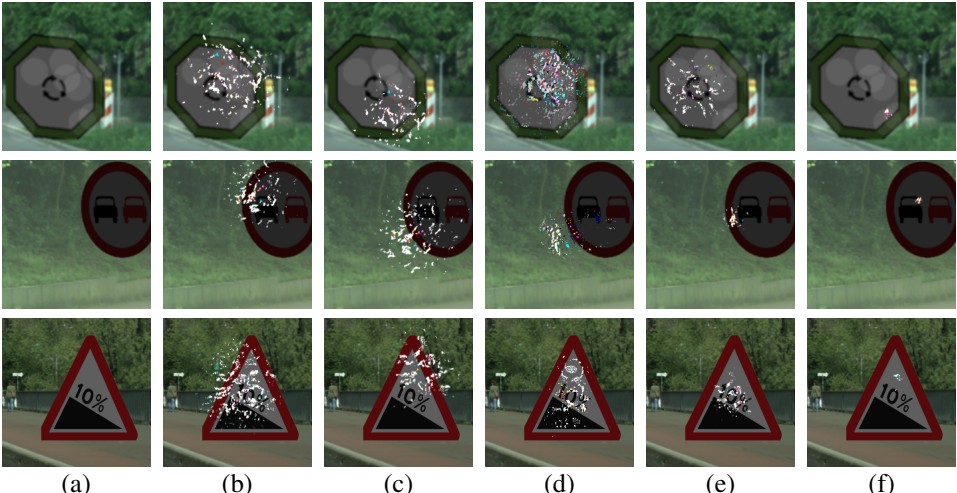

| (a) | (b) | (c) | (d) | (e) | (f) |

Figure 6: Examples of recovered triggers overlaid on clean images. From left to right: **(a)** clean image, **(b)** triggers recovered by (Wang et al., 2019), **(c)** triggers recovered by (Liu et al., 2019), **(d)** triggers recovered by (Guo et al., 2019), **(e)** triggers recovered by our method without topological prior, and **(f)** triggers recovered by our method with topological prior.

the detection network by optimizing cross entropy loss using the Adam optimizer (Kingma & Ba, 2014). The hidden state size, number of layers of $MLP_\alpha$, $MLP_\beta$, as well as optimizer learning rate, weight decay and number of epochs are optimized using Bayesian hyperparameter search[4] for 500 rounds on 8-fold cross-validation.

**Evaluation metrics**: We follow the settings in (Sikka et al., 2020). We report the mean and standard deviation of two metrics: area under the ROC curve (AUC) and accuracy (ACC). Specifically, we evaluate our approach on the whole set by doing an 8-fold cross validation. For each fold, we use 80% of the models for training, 10% for validation, and the rest 10% for testing.

**Results**: Tables 1 and 2 show the quantitative results on the Trojaned-MNIST/CIFAR10 and TrojAI datasets, respectively. The reported performances of baselines are reproduced using source codes provided by the authors or quoted from related papers. The best performing numbers are highlighted in bold. From Tab. 1 and 2, we observe that our method performs substantially better than the baselines. It is also worth noting that, compared with these baselines, our proposed method extracts fix-sized features for each model, independent of the number of classes, architectures, trigger types, etc. By using the extracted features, we are able to train a separate Trojan detection network, which is salable and model-agnostic.

Table 2: Performance comparison on the TrojAI dataset.

| Method | Metric | TrojAI-Round1 | TrojAI-Round2 | TrojAI-Round3 | TrojAI-Round4 |
|---|---|---|---|---|---|
| NC | AUC | $0.50 \pm 0.03$ | $0.63 \pm 0.04$ | $0.61 \pm 0.06$ | $0.58 \pm 0.05$ |
| ABS | AUC | $0.68 \pm 0.05$ | $0.61 \pm 0.06$ | $0.57 \pm 0.04$ | $0.53 \pm 0.06$ |
| TABOR | AUC | $0.71 \pm 0.04$ | $0.66 \pm 0.07$ | $0.50 \pm 0.07$ | $0.52 \pm 0.04$ |
| ULP | AUC | $0.55 \pm 0.06$ | $0.48 \pm 0.02$ | $0.53 \pm 0.06$ | $0.54 \pm 0.02$ |
| DLTND | AUC | $0.61 \pm 0.07$ | $0.58 \pm 0.04$ | $0.62 \pm 0.07$ | $0.56 \pm 0.05$ |
| Ours | AUC | $\mathbf{0.90 \pm 0.02}$ | $\mathbf{0.87 \pm 0.05}$ | $\mathbf{0.89 \pm 0.04}$ | $\mathbf{0.92 \pm 0.06}$ |
| NC | ACC | $0.53 \pm 0.04$ | $0.49 \pm 0.02$ | $0.59 \pm 0.07$ | $0.60 \pm 0.04$ |
| ABS | ACC | $0.70 \pm 0.04$ | $0.59 \pm 0.05$ | $0.56 \pm 0.03$ | $0.51 \pm 0.05$ |
| TABOR | ACC | $0.70 \pm 0.03$ | $0.68 \pm 0.08$ | $0.51 \pm 0.05$ | $0.55 \pm 0.06$ |
| ULP | ACC | $0.58 \pm 0.07$ | $0.51 \pm 0.03$ | $0.56 \pm 0.04$ | $0.57 \pm 0.04$ |
| DLTND | ACC | $0.59 \pm 0.04$ | $0.61 \pm 0.05$ | $0.65 \pm 0.04$ | $0.59 \pm 0.06$ |
| Ours | ACC | $\mathbf{0.91 \pm 0.03}$ | $\mathbf{0.89 \pm 0.04}$ | $\mathbf{0.90 \pm 0.03}$ | $\mathbf{0.91 \pm 0.04}$ |

Fig. 6 shows a few examples of recovered triggers. We observe that, compared with the baselines, the triggers found by our method are more compact and of better quality. This is mainly due to the introduction of topological constraints. The improved quality of recovered triggers directly results in improved performance of Trojan detection.

---

[4]https://github.com/hyperopt/hyperopt

**Ablation study of loss weights**: For the loss weights $\lambda_1$ and $\lambda_2$, we empirically choose the weights which make reverse engineering converge the fastest. This is a reasonable choice as in practice, time is one major concern for reverse engineering pipelines.

Despite the seemingly ad hoc choice, we have observed that our performances are quite robust to all these loss weights. As topological loss is a major contribution of this paper, we conduct an ablation study in terms of its weight ($\lambda_2$) on TrojAI-Round4 dataset. The results are reported in Fig. 7. We observe that the proposed method is quite robust to $\lambda_2$, and when $\lambda = 10$, it achieves slightly better performance (AUC: $0.92 \pm 0.06$) than other choices.

**Ablation study of number of training model samples**: The trigger features and Trojan detection network are important in achieving SOTA performance. To further demonstrate the efficacy of the proposed diversity and topological loss terms, we conduct another ablation study to investigate the case with less training model samples, and thus a weaker Trojan detection network.

Figure 7: Ablation study results for $\lambda_2$.

The ablation study in terms of number of training samples on TrojAI-Round4 data is illustrated in Tab. 3. We observe that the proposed topological loss and diversity loss will boost the performance with/without a fully trained Trojan-detection network. These two losses improve the quality of the recovered trigger, in spite of how the trigger information is used. Thus even with a limited number of training samples (e.g., 25), the proposed method could still achieve significantly better performance than the baselines.

**Ablation study for loss terms**: We investigate the individual contribution of different loss terms used to search for the latent triggers. Tab. 4 lists the corresponding performance on the TrojAI-Round4 dataset. We observe a decrease in AUC (from 0.92 to 0.89) if the topological loss is removed. This drop is expected as the topological loss helps to find more compact triggers. Also, the performance drops signifi-

Table 3: Ablation study for # of training samples.

| # of samples | Ours | w/o topo | w/o diversity |
|---|---|---|---|
| 25 | $\mathbf{0.77 \pm 0.04}$ | $0.73 \pm 0.03$ | $0.68 \pm 0.04$ |
| 50 | $\mathbf{0.81 \pm 0.03}$ | $0.76 \pm 0.05$ | $0.73 \pm 0.02$ |
| 100 | $\mathbf{0.84 \pm 0.05}$ | $0.78 \pm 0.06$ | $0.76 \pm 0.03$ |
| 200 | $\mathbf{0.86 \pm 0.04}$ | $0.82 \pm 0.04$ | $0.79 \pm 0.05$ |
| 400 | $\mathbf{0.90 \pm 0.05}$ | $0.85 \pm 0.03$ | $0.82 \pm 0.04$ |
| 800 | $\mathbf{0.92 \pm 0.06}$ | $0.89 \pm 0.04$ | $0.85 \pm 0.02$ |

cantly (from 0.92 to 0.85 in AUC) if the diversity loss is removed. We also report the performance by setting $N_T = 2$; when $N_T = 2$, the performance increases from 0.85 to 0.89 in AUC. The reason is that with diversity loss, we are able to generate multiple diverse trigger candidates, which increases the probability of recovering the true trigger when the target class is unknown. Our ablation study justifies the use of both diversity and topological losses.

In practice, we found that topological loss can improve the convergence of trigger search. Without topological loss, it takes $\approx 50$ iterations to find a reasonable trigger (Fig. 6(e)). In contrast, with the topological loss, it takes only $\approx 30$ iterations to converge to a better recovered trigger (Fig. 6(f)). The rationale

Table 4: Ablation results of loss terms.

| Method | TrojAI-Round4 |
|---|---|
| w/o topological loss | $0.89 \pm 0.04$ |
| w/o diversity loss ($N_T = 1$) | $0.85 \pm 0.02$ |
| $N_T = 2$ | $0.89 \pm 0.05$ |
| with all loss terms ($N_T = 3$) | $\mathbf{0.92 \pm 0.06}$ |

is that, as the topological loss imposes strong constraints on the number of connected components, it largely reduces the search space of triggers, consequently, making the convergence of trigger search much faster. This is worth further investigation.

**Unsupervised vs supervised**: Our technical contributions are agnostic of whether the detection is supervised (i.e., using annotated models) or unsupervised. The proposed diversity and topological losses are used to improve the quality of a reconstructed trigger, using only a single model and input data. More discussions and results in terms of (un)supervised settings are included in Sec. A.5.

## 5 CONCLUSION

In this paper, we propose a diversity loss and a topological prior to improve the quality of the trigger reverse engineering for Trojan detection. These loss terms help finding high quality triggers efficiently. They also avoid the dependant of the method to the target label. On both synthetic datasets and publicly available TrojAI benchmarks, our approach recovers high quality triggers and achieves SOTA Trojan detection performance.

**Ethics Statement:** As we have developed better Trojan detection algorithm and introduce the method in details, the attackers may inversely create Trojaned models that are more difficult to detect based on the limitations of current method. Attack and defense will always coexist, which pushes researchers to keep developing more efficient algorithms.

**Reproducibility Statement:** The implementation details are mentioned in Sec. 4. The details of the data are provided in Sec. A.2 of Appendix. The details of Trojan detection classifier are described in Sec. A.1 of Appendix. The used computation resources are specified in Sec. A.7 of Appendix.

**Acknowledgement:** The authors thank anonymous reviewers for their constructive feedback. This effort was partially supported by the Intelligence Advanced Research Projects Agency (IARPA) under the contract W911NF20C0038. The content of this paper does not necessarily reflect the position or the policy of the Government, and no official endorsement should be inferred.

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

# A APPENDIX

## A.1 LEARNING-BASED TROJAN-DETECTION NETWORK

While bottom-up trigger generation uses appearance heuristics to search for possible triggers, we cannot guarantee the recovered triggers are true triggers, even for Trojaned models. Many other perturbations can create label flipping effects, such as adversarial samples and modifications specific on the most semantically critical region. See Figure 8 for illustrations. These examples can easily become false positives for a Trojan detector; they can be good trigger candidate generated by the reverse engineer pipeline.

To fully address these issues, we propose a top-down Trojan detector learned using clean and Trojaned models. It helps separate true Trojan triggers from other false positives. To this end, we propose to extract features from the reverse engineered Trojan triggers and train a separate shallow neural network for Trojan detection.

For each model, we generate a diverse set of $N_T$ possible triggers for each of its $K$ output classes to scan for possible triggers. As described above, we extract one feature for each generated trigger. As a result, for each model we have $N_T \times K$ sets of features.

**Regularizer**: The regularizer $R(\mathbf{m})$ consists of a mass term and a size term. For mass, we use $\bar{\mathbf{m}}$ as the average value of $\mathbf{m}$. For size, we normalize the mask $\mathbf{m}$ into a distribution $p(x, y)$ over $x$ and $y$. To capture the spatial extent of mask $m$, we compute the standard deviation of $X \sim p(x)$ as $\delta_X$ and $Y \sim p(y)$ as $\delta_Y$. As a result, $R(\mathbf{m}) = \bar{\mathbf{m}} + \delta_X + \delta_Y$ is the regularizer.

**Classification network**: After the features are extracted, we build a neural network for Trojan detection, which takes as input for a given image classifier, the bag of features of its generated triggers and outputs a scalar score of whether the model is Trojaned or not.

Since different models may have different numbers of output classes $K$, the number of features varies for each model. Therefore, a sequence of modeling architectures – such as bag-of-words, Recurrent Neural Networks and Transformers – could be employed to aggregate the $N_T \times K$ features into a $0/1$ Trojan classification output.

Figure 8: Our Trojan detection method combines bottom-up Trigger reverse engineering under topological constraints, with top-down classification. Such a combination allows us to accurately isolate Trojan triggers from non-Trojan patterns such as adversarial noise and object modifications.

Given a set of annotated models, clean or infected, supervised learning can be applied to train the classification network. Empirically, we found that a simple bag-of-words technique achieved the best Trojan detection performance while also being fast to run and data efficient. Specifically, let the bag of features be $\{v_i\}$, $i = 1, \ldots, N_T K$. The features are first transformed individually using an MLP, followed by average pooling across the features and another MLP to output the Trojan classification:

$$\vec{h}_i = MLP_\alpha(\vec{v}_i), \quad \vec{h} = \frac{1}{N_T K} \sum_i \vec{h}_i, \quad s = MLP_\beta(\vec{h}), \quad i = 1, \ldots N_T K. \tag{7}$$

## A.2 DETAILS ABOUT THE TROJAI DATASETS

**TrojAI-Round1, Round2, Round3, Round4 datasets**: These datasets are provided by US IARPA/NIST[5] and contain trained models for traffic sign classification (for each round, 50% of total models are Trojaned). All the models are trained on synthetically created image data of non-real traffic signs superimposed on road background scenes. Trojan detection is more difficult on Round2/Round3/Round4 compared to Round1 due to following reasons:

---

[5]https://pages.nist.gov/trojai/docs/data.html

- Round2/Round3 have more number of classes: Round1 has 5 classes while Round2/Round3 have 5-25 classes.

- Round2/Round3 have more trigger types: Round1 only has polygon triggers, while Round2/Round3 have both polygon and Instagram filter based triggers.

- The number of source classes are different: all classes are poisoned in Round1, while 1, 2, or all classes are poisoned in Round2/Round3.

- Round2/Round3 have more type of model architectures: Round1 has 3 architectures, while Round2/Round3 have 23 architectures.

Round3 experimental design is identical to Round2 with the addition of Adversarial Training. Two different Adversarial Training approaches: Projected Gradient Descent (PGD), Fast is Better than Free (FBF) (Wong et al., 2019) are used.

Unlike the previous rounds, Round4 can have multiple concurrent triggers. Additionally, triggers can have conditions attached to their firing. The differences are listed as follows:

- All triggers in Round4 are one to one mappings, which means a trigger flips a single source class to a single target class.

- Three possible conditionals, spatial, spectral and class are attached to triggers within this dataset.

- Round4 has remove the very large model architectures to reduce the training time.

Round1, Round2, Round3 and Round4 have 1000, 1104, 1008 and 1008 models respectively.

### A.3    SUPPORTING ADDITIONAL TRIGGER CLASSES

Different classes of Trojan triggers are being actively explored in Trojan detection benchmarks. For image classification, the TrojAI datasets include localized triggers which can be directly applied to objects along with global filter-based Trojans, where the idea is that a color filter could be attached to the lens of camera and results in a global image transformation.

Our top-down bottom-up Trojan detection framework is designed to support multiple classes of Trojan triggers, where each trigger class gets its dedicated reverse engineering approach and pathway in the Trojan detection network. Adding support to a new class of triggers, e.g. color filters, amounts to adding a reverse engineering approach for color filters and adding a appearance feature descriptor for Trojan classification.

**Reverse engineering color filter triggers.**    The pipeline of reverse engineering color filter triggers is illustrated in Fig. 9. Comparing to reverse engineering local triggers in Fig.4, the filter editor and the loss functions are adjusted to find color filter triggers.

We model a color filter trigger using a per-pixel position-dependent color transformation. Let $[r_{ij}, g_{ij}, b_{ij}]$ be the color of a pixel of input image $\mathbf{x}$ at location $(i, j)$, and we model a color filter trigger using an MLP $E(\cdot; \theta^{\text{filter}})$ with parameters $\theta^{\text{filter}}$ which performs position-dependent color mapping:

$$[\hat{r_{ij}}, \hat{g_{ij}}, \hat{b_{ij}}] = E([r_{ij}, g_{ij}, b_{ij}, i, j]; \theta^{\text{filter}}). \tag{8}$$

Here $[\hat{r_{ij}}, \hat{g_{ij}}, \hat{b_{ij}}]$ is the color of pixel $(i, j)$ of the triggered image $\hat{\mathbf{x}}$. For TrojAI datasets we use a 2-layer 16 hidden neurons MLP to model the color filters, as it learns sufficiently complex color transforms while being fast to run:

$$L^{\text{filter}} = L_{flip}^{\text{filter}} + \lambda_1^{\text{filter}} L_{div}^{\text{filter}} + \lambda_2^{\text{filter}} R^{\text{filter}}(\theta^{\text{filter}}). \tag{9}$$

The label flipping loss $L_{flip}^{\text{filter}}$ remains identical:

$$L_{flip}^{\text{filter}} = \hat{y}_c. \tag{10}$$

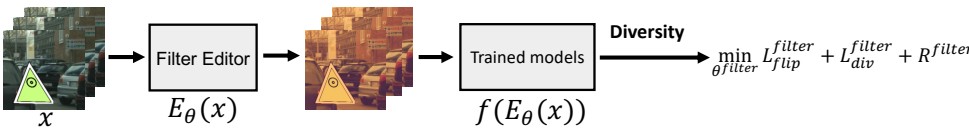

Figure 9: Reverse engineering of global color filter triggers.

The diversity loss $L_{div}^{\text{filter}}$ is designed to induce diverse color transforms. We use how a color filter transforms 16 random $(r, g, b, i, j)$ tuples to characterize a color filter $E(\cdot; \theta^{\text{filter}})$. We record the 16 output $(\hat{r}, \hat{g}, \hat{b})$ tuples as a 48-dim descriptor of the color filter, denoted as $u_{\theta^{\text{filter}}}$. The diversity loss is the $L_2$ norm of the current candidate to previously found triggers:

$$L_{div}^{\text{filter}} = -\sum_{j=1}^{N_T}\sum_{i=1}^{j-1} ||u_{\theta_i^{\text{filter}}} - u_{\theta_j^{\text{filter}}}||_2. \tag{11}$$

The regularizer term $R^{\text{filter}}(\theta^{\text{filter}})$ is simply an L2 regularizer on the MLP parameters to reduce the complexity on the color transforms:

$$R^{\text{filter}}(\theta^{\text{filter}}) = ||\theta^{\text{filter}}||_2^2. \tag{12}$$

**Feature extractor for color filter triggers.** For each model we use reverse engineering to generate $K$ classes by $N_T^{filter}$ diverse color filter triggers. For each color filter trigger, we combine an appearance descriptor with label flipping loss $L_{flip}^{\text{filter}}$ and diversity loss $L_{div}^{\text{filter}}$ as its combined feature descriptor. For the appearance descriptor, we use the same descriptor discussed in diversity: how a color filter $E(\cdot; \theta^{\text{filter}})$ transforms 16 random $(r, g, b, i, j)$ tuples. We record the 16 output $(\hat{r}, \hat{g}, \hat{b})$ tuples as a 48-dim appearance descriptor.

As a result for each model we have $N_T^{filter}K$ features for color filter triggers.

**Trojan classifier with color filter triggers.** A bag-of-words model is used to aggregate those features. The aggregated features across multiple trigger classes, e.g. color filters and local triggers, are concatenated and fed through an MLP for final Trojan classification as below:

$$\vec{h}_i^{filter} = MLP_\alpha^{filter}(\vec{v}_i^{filter}), \quad \vec{h}^{filter} = \frac{1}{N_T^{filter}K}\sum_i \vec{h}_i^{filter}, \quad i = 1, \dots N_T^{filter}K \tag{13}$$

$$\vec{h}_i^{local} = MLP_\alpha^{local}(\vec{v}_i^{local}), \quad \vec{h}^{local} = \frac{1}{N_TK}\sum_i \vec{h}_i^{local}, \quad i = 1, \dots N_TK \tag{14}$$

$$s = MLP_\beta([\vec{h}^{filter}; \vec{h}^{local}]), \tag{15}$$

For the TrojAI datasets, color filter reverse engineering is conducted using Adam optimizer with learning rate $3 \times 10^{-2}$ for 10 iterations. Hyperparameters are set to $\lambda_1^{\text{filter}} = 0.05$ and $\lambda_2^{\text{filter}} = 10^{-4}$. We also set $N_T = 2$ and $N_T^{filter} = 8$.

A.4 COMPARED WITH ADDITIONAL SOTA METHOD

We also compare our proposed method with another parallel work (Shen et al., 2021). Here we directly quote the numbers (Accuracy) from the paper for comparison. Note that the eval protocols are different, but only subtly. Additionally, we provide the comparison of recovered triggers with (Shen et al., 2021) in Fig. 10 to demonstrate that the proposed method could also improve the quality of recovered triggers.

h

Table 5: Comparison with (Shen et al., 2021).

| Method | Round1 | Round2 | Round3 | Round4 |
|---|---|---|---|---|
| Shen et al. (2021) | 0.90 | **0.89** | **0.91** | 0.89 |
| Ours | **0.91** | **0.89** | 0.90 | **0.91** |

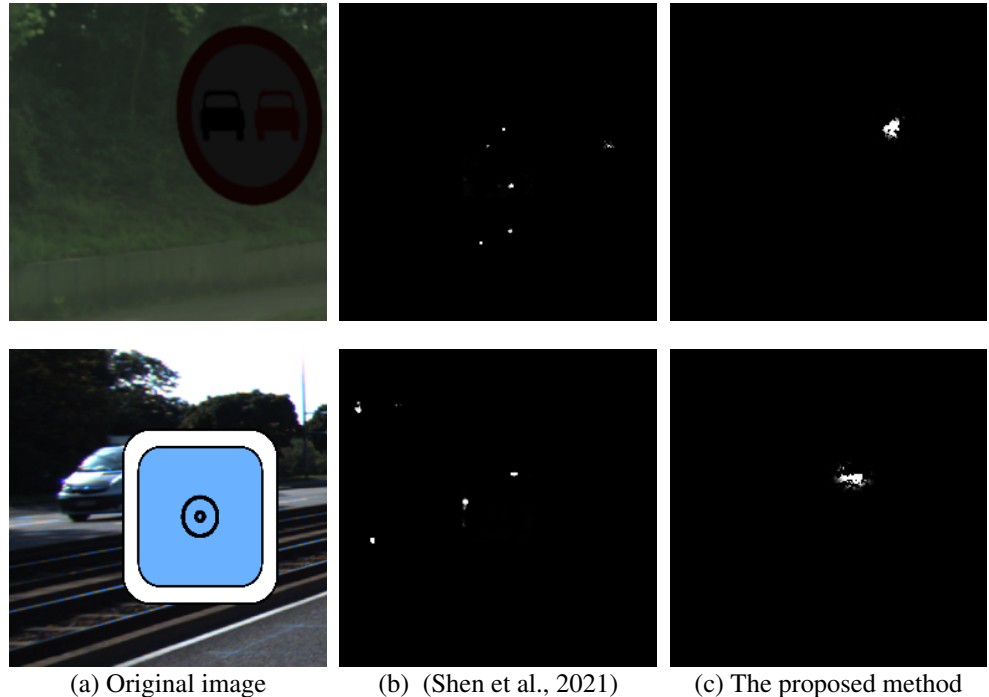

| (a) Original image | (b) (Shen et al., 2021) | (c) The proposed method |

Figure 10: Recovered triggers compared with (Shen et al., 2021)

### A.5 UNSUPERVISED SETTING FOR TROJAN DETECTION

Indeed, our method outperforms existing methods in different settings: fully supervised setting with annotated models, and unsupervised setting. In the main text (Tab. 3), we have already demonstrated that with a limited number of annotated models, our method outperformed others. To make a fair comparison, following Neural Cleanse and DLTND, we also use the simple technique based on Median Absolute Deviation (MAD) for trojan detection. We report the performance of Round 4 data of TrojAI in Tab. 6.

Table 6: Unsupervised performances on Trojan.

| Method | AUC | ACC |
|---|---|---|
| NC | 0.58 | 0.60 |
| ABS | 0.53 | 0.51 |
| TABOR | 0.52 | 0.55 |
| ULP | 0.54 | 0.57 |
| DLTND | 0.56 | 0.59 |
| Ours | **0.63** | **0.65** |

Because of the better trigger quality, due to the proposed losses, our method outperforms baselines such as Neural Cleanse. We also note that, in Tab. 6, all methods (including ours) perform unsatisfactorily in the unsupervised setting. This brings us back to the discussion as to whether a supervised setting is justified in Trojan detection (although this is not directly relevant to our method).

From the research point of view, we believe that data-driven methods for Trojan detection are unavoidable as the attack techniques continue to develop. Like in many other security research problems, Trojan attack and defense are two sides of the same problem that are supposed to advance together. When the problem was first studied, classic unsupervised methods such as Neural Cleanse are sufficient. In recent years, the attack techniques have continued to develop, exploiting the entire dataset and leveraging techniques like adversarial training. Meanwhile, detection methods are

confined with only the given model and a few sample data. For defense methods to move forward and to catch up with the attack methods, it seems only natural and necessary to exploit supervised approaches, e.g., learning patterns from public datasets such as the TrojAI benchmarks.

## A.6    IMPLEMENTATION OF BASELINES

We carefully choose the methods with available codes from the authors as our baselines and follow the instructions to obtain the reported results. And we list all the available repositories here:

Neural Cleanse: https://github.com/bolunwang/backdoor

ABS: https://github.com/naiyeleo/ABS

TABOR: https://github.com/UsmannK/TABOR

ULP: https://github.com/UMBCvision/Universal-Litmus-Patterns

DLTND: https://github.com/wangren09/TrojanNetDetector

Part of the reason for the strong benefit of our method, as well as the ablated version (without the new losses), is because of the availability of the annotated training models.

## A.7    COMPUTATION RESOURCES

We implement our method on a server with an Intel(R) Xeon(R) Gold 6140 CPU @ 2.30GHz and 1 Tesla V100 GPUs (32GB Memory).

