# OpenReview forum: "Trigger Hunting with a Topological Prior for Trojan Detection"
_ICLR.cc/2022/Conference — ICLR 2022 Poster_

### Official Review · Reviewer_momo · 2021-10-26

**Correctness:** 3
**Technical Novelty And Significance:** 3
**Empirical Novelty And Significance:** 3
**Recommendation:** 5
**Confidence:** 5

**Main Review:**

Strengths:
1. The experiments show fantastic Trojan detection accuracy under the TrojAI benchmark settings.
2. The idea of the diversity penalty is simple and straightforward. I like the consideration of unknown target labels and multiple triggers, which may be the closest to the real scenes.
3. A very detailed ablation study for the contribution of each item.

Weaknesses:

1. It took me a while to understand the idea of topological simplicity and why such prior exists in the trojan attack settings. The motivation needs to be further clarified. For example, can the embedded trigger be a set of scattering points of pixels? If can, will the topological simplicity exist? This also leads to another question about the quality evaluation of recovered triggers. In fig 6, It is doubtful to consider the “more compact” as a good metric for triggers.
2. The second limitation of the proposed method is the training of the detection network, which needs annotation of the trojan model. To my knowledge, a very unique characteristic of a trojan attack is its stealthiness. Once the type and method of attack are exposed publicly, the threat does not exist. Are there any possible ways to apply the proposed method in a more realistic scene without the annotations of the trojaned model?
3. It will be better if the authors provide a more detailed implementation of baselines since the performance increase is very astonishing. Also, a clear analysis of the impact of trigger types is better to be exhibited.

**Summary Of The Paper:**

This paper presents a novel method for trigger detection against trojan attacks (especially the TrojAI benchmark). The key idea is to introduce the diversity penalty and the topological simplicity to help find more different high-quality triggers in the image domain.

**Summary Of The Review:**

This paper is well-written and proposes a pretty novel trigger hunting method. However, the limitation of this method is also very obvious, not good enough for real threats. I will increase my score if the authors can provide more insights under more general settings of trojan attacks except for only TrojAI benchmarks.

---

> ### Author Response · Authors · 2021-11-23
> **Response to the comments of Reviewer momo**
>
> Thanks for your constructive feedback. Please refer to the **General response** section for discussions regarding the scattered trigger concern, and unsupervised setting. Below we address remaining concerns one-by-one.
>
> **Q1**: It will be better if the authors provide a more detailed implementation of baselines since the performance increase is very astonishing.
>
> **A**: We carefully choose the methods with available codes from the authors as our baselines and follow the instructions to obtain the reported results. And we list all the available repositories here:
>
> Neural Cleanse: https://github.com/bolunwang/backdoor
>
> ABS: https://github.com/naiyeleo/ABS
>
> TABOR: https://github.com/UsmannK/TABOR
>
> ULP: https://github.com/UMBCvision/Universal-Litmus-Patterns
>
> DLTND: https://github.com/wangren09/TrojanNetDetector
>
> Part of the reason for the strong benefit of our method, as well as the ablated version (without the new losses), is because of the availability of the annotated training models. For detailed discussion, please refer to General response Q1.
>
> **Q2**: Also, a clear analysis of the impact of trigger types is better to be exhibited.
>
> **A**: Thanks for the suggestions. We will add more detailed analysis regarding different trigger sizes and shapes in the final version. Please do not hesitate to let us know if we misunderstood your question.
>
> **Q3**: The authors can provide more insights under more general settings of trojan attacks except for only TrojAI benchmarks.
>
> **A**: In our experiments, we also provided data created with CIFAR10 and MNIST. Hope this provides some reassurance that the method is not 'overfitting' to TrojAI benchmarks. We are confident about the generalizability of our method because the approach is very general and is not dataset-specific.

---

> > ### Author Response · Authors · 2021-12-01
> > **Follow up**
> >
> > Dear reviewer,
> >
> > Do you still have any concerns about our manuscript? We are sincerely looking forward to your further feedback!

---

> > ### Comment · Reviewer_momo · 2021-12-02
> > **Thanks for your response**
> >
> > Thanks for responding to my comments.
> >
> > The response has addressed most of my concerns. Thus I changed my score to 6.

---

> > > ### Author Response · Authors · 2021-12-02
> > > **Thanks for your feedback**
> > >
> > > Dear reviewer,
> > >
> > > Thanks very much for your feedback! We really appreciate your going through our response and agreeing to update your recommendation. However, looking at your initial review, we still couldn't see your score being updated. Would you please kindly take a look?
> > >
> > > Sincerely,
> > >
> > > Authors

---

### Official Review · Reviewer_6VNg · 2021-10-31

**Correctness:** 3
**Technical Novelty And Significance:** 3
**Empirical Novelty And Significance:** 3
**Recommendation:** 8
**Confidence:** 3

**Main Review:**

### Strengths and Weaknesses:

+ The paper is very well-written.
It flows well and provides extensive explanations on abstract topics such as persistent homology to help the reader understand them easily.

+ The proposed approach seems novel and promising.
The experimental results are comprehensive, and are done on synthetic (MNIST and CIFAR-10) as well as real-world (TrojAI) backdoored models.
It is shown that using the proposed approach one can get significant improvements over state-of-the-art baselines such as Neural Cleanse [[1](https://ieeexplore.ieee.org/abstract/document/8835365)] and DL-TND [[2](https://arxiv.org/abs/2007.15802)].

+ Although the use of topological loss seems to be enhancing the detection of backdoor models, an explanation as to why this choice is valid is somehow missing.
It is pointed out that this way the topological loss would limit the search space of all possible triggers, which as a result would increase the chances of finding the true one.
While this makes total sense, it also brings a natural question to mind: are all the possible triggers compact?
In other words, how does the introduced topological loss behave in case the adversary poisons the model with a scattered trigger?
I think this question needs to be answered to help the readers understand why a topological loss needs to be employed.

### Minor Comments/Questions:

+ To minimize Eq. (2), a sum over all the validation samples is taken, or they are minimized separately? If it is the latter, please add a summation to indicate this.

+ In Table 3, the number of samples means the number of models in the training data or the number of validation samples used for reverse engineering?

+ In the assumptions of Section 3.1 $\mathbf{x}$ needs to be in $\mathbb{R}^{3\times M \times N}$ to accommodate the three RGB channels.

**Summary Of The Paper:**

This paper presents a method for the detection of backdoored neural networks.
It is assumed that the user only will have access to the trained model and a few clean validation samples, not the training data.
As such, a reverse engineering approach is taken to solve the task: i.e. using the validation samples, the method aims to recover possible backdoor triggers.
To this end, a loss function with three terms is minimized:

1. a _label flipping_ loss to extract triggers that can subvert the classifier's decision,
2. a _diversity_ loss that aims to generate various triggers with different patterns, and
3. a _topological_ loss that tries to encourage the compactness of the recovered triggers.

Based on these loss terms, triggers are reverse engineered for a dataset containing malicious and benign models.
Then, using these triggers a set of features are built for each model (Trojaned vs. benign).
Using these features, a binary classifier is trained to predict if the models are backdoored or not.
The performance of the proposed approach is tested under various experimental settings and ablation studies.

**Summary Of The Review:**

This work introduces a novel approach for the detection of backdoor neural networks.
The performance of the approach seems to be promising.
However, there are some speculations around the use of topological loss for detection as the compactness of the triggers is not a necessity in backdoor attacks.
If the authors can elaborate on this assumption, I would be happy to increase my score.

---

> ### Author Response · Authors · 2021-11-23
> **Response to the comments of Reviewer 6VNg**
>
> Thanks for your constructive feedback.  Please refer to the **General response Q2** for discussions regarding the scattered trigger concern. Below we address minor presentation issues.
>
> **Q1**: To minimize Eq. (2), a sum over all the validation samples is taken, or they are minimized separately?
>
> **A**: Eq.(2) is minimized for each sample to obtain the features for each sample/model. We will clarify this in the revised version.
>
> **Q2**: In Table 3, the number of samples means the number of models in the training data or the number of validation samples used for reverse engineering?
>
> **A**: As indicated by the title of Table.3, it’s the number of samples in the training data.
>
> **Q3**: In the assumptions of Section 3.1 **x** needs to be in $\mathbb{R}^{3\times M \times N}$  to accommodate the three RGB channels.
>
> **A**: Thanks for your suggestion, and we’ve modified it as you suggested. The changes are colored in blue in the main text.

---

### Official Review · Reviewer_Tmo4 · 2021-10-31

**Correctness:** 2
**Technical Novelty And Significance:** 3
**Empirical Novelty And Significance:** 2
**Recommendation:** 5
**Confidence:** 4

**Main Review:**

The introduction of the diversity loss and topological loss is interesting and seems to improve the quality of generated triggers. The evaluation is conducted on a large benchmark. I have the following concerns with the paper.

1. The current approach seems to heavily rely on a large set of models with the same or closely related task. A reasonable detection result (>0.8 AUC) requires at least 50 models as shown in Table 3. This does not seem practical. How can a defender acquire such a large set of clean models? Even if the defender can obtain a set of clean models, the task for these model also need to be similar, e.g., traffic sign detection in TrojAI datasets. What if these models have different tasks? Can the proposed approach still work? The paper needs to be better motivated why having a large set of clean models with similar task is practical and realistic.

2. Following up the first point, the comparison with several baselines does not seem fair. NC, ABS, and TABOR do not require any clean models for training. The detection is directly carried out on a given model (either benign or trojaned). The proposed approach relies on the assumption of having a set of clean models, which has a different threat model than these baselines. A fair comparison would be some modified version of NC, ABS or TABOR that also takes a set of clean models into consideration during detection. For instance, the defender could first run these baselines on the clean set and get an estimation of how clean models would behave. She then uses this information to guide the detection.

3. A recent state-of-the-art approach (Shen et al.) is not compared. The paper only briefly mentions it in the related work but does not empirically compare with it. The results from the original paper (Shen et al.) are very similar with marginally differences. The experiment of the proposed approach is only evaluated on 10% of the TrojAI datasets as it requires training and validating on the remaining 90%, whereas Shen et al. evaluated on the whole set. The proposed approach can be better compared with the baseline by presenting the results from TrojAI test set that is evaluated remotely by the TrojAI organizer. This can rule out the possibility that the approach does not overfit on the local training set as this is a training based method.

4. There is no discussion and evaluation on adaptive attacks. When knowing the existence of the proposed detection, the attacker can trojan a model by maximizing the mask using the losses in Equation (2). She can also add those reverse engineered triggers in the training set during poisoning to counter the effect of generating a set of small triggers. The trojan detection network can also be considered by incorporating in the loss function.

5. The paper does not clearly separate the contribution of existing works and this paper. For instance, the trigger reverse engineering technique by using a mask and a pattern in Section 3.1 is proposed by NC. Utilizing continuous values instead of binary values for the mask is also proposed by NC. This should either be discussed in a section regarding existing works or explicitly credits the contribution to existing works.

6. From the ablation results in Table 3, without the topological loss or the diversity loss, the results are already better than evaluated baselines. Why is this the case? What are other factors contributing to the final good detection results? Without the above two losses, it is more or less NC, which should have a very low detection result. Do the author care to explain in detail which component of the approach leads to the good results.

7. The presentation needs improvement. Equation (2) introduces three losses, namely, the label-flipping loss, diversity loss, and topological loss. The first two losses are discussed in Section 3.1, while the topological loss is presented in a separate section, Section 3.2. These three losses should be at the same section level. The caption of Figure 5 is hard to be distinguished from the main text. There are no images with the injected triggers in Figure 6.

**Summary Of The Paper:**

This paper proposes a trojan detection method using reverse engineering techniques. It improves existing approaches by incorporating a trigger diversity loss and a topological loss during trigger generation. The diversity loss aims to generate a set of triggers with different patterns and locations. The topological loss is to make triggers more connected with fewer perturbations spread out on the input. The paper builds a neural network to detect trojaned models by using the features of generated triggers such as the number of trigger pixels, the mean and standard deviation of triggers in horizontal and vertical directions, etc. This paper evaluates the proposed approach on two synthetic datasets and TrojAI benchmarks.

**Summary Of The Review:**

1. The setup does not seem practical.
2. The comparison with existing works is unfair.
3. There is no comparison with a state-of-the-art approach.
4. The paper is not evaluated on adaptive attacks.
5. It is not clear about the contribution of different components.
6. The presentation of the paper needs improvements.

---

> ### Author Response · Authors · 2021-11-23
> **Response to the comments of Reviewer Tmo4**
>
> Thanks for your constructive feedback. Please refer to the **General response Q1** for discussions regarding supervised/unsupervised settings and the comparison results. Below we address remaining concerns one-by-one.
>
> **Q1**: What if these models have different tasks? Can the proposed approach still work?
>
> **A**: Our approach is as adaptive to different tasks as a standard reverse engineering approach. A reverse engineering method relies on the relationship between the model output and model input. In principle it can apply to any single-output model (classification or regression).
>
> **Q2**: The experiment of the proposed approach is only evaluated on 10% of the TrojAI datasets as it requires training and validating on the remaining 90%, whereas Shen et al. evaluated on the whole set.
>
> **A**: Sorry for the misunderstanding about the evaluation metric. Actually,  our approach is evaluated by doing an 8-fold cross validation, and for each fold, we train on 90% and evaluate on 10%. In that way almost the whole set was used for eval. We have clarified this in the revised version, and please find it in the ‘Evaluation metrics’ part of Page 8 in the main text. The changes in the main text are colored in blue.
>
> **Q3**: The proposed approach can be better compared with the baseline by presenting the results from TrojAI test set that is evaluated remotely by the TrojAI organizer.
>
> **A**: We did evaluate the preliminary version of our method on the TrojAI competition. It achieved top results across rounds 1-4 of the competition. When our method was finalized, the competitions of rounds 1-4 were closed. We can only evaluate the final version of our method on the released training set and report cross validation results (Tables 2).
>
> **Q4**: Comparison with SOTA (Shen et al. 2021).
>
> **A**: We’d like to stipulate that (Shen et al. 2021) is parallel to us. Here we directly quote the numbers (Accuracy) from the paper for comparison. As clarified in Q4, the eval protocols are different, but only subtly.
>
> | Method | Round1 | Round2 | Round3 |Round4 |
> |:------:|:------:|:------:|:------:|:------:|
> |  (Shen et al.) | 0.90 |   **0.89**  |   **0.91** | 0.89 |
> | Ours  |  **0.91**  |  **0.89** | 0.90 | **0.91**|
>
> We also note that the main contribution of our paper is the quality of the recovered triggers. To this end, we provide qualitative examples to show that our method recovers triggers that are much more reasonable than (Shen et al.2021). We use the codes provided by the authors at https://github.com/PurduePAML/K-ARM_Backdoor_Optimization to generate the trigger. See Figure 10 in the Appendix of the revision. Triggers generated by our method are much more compact than triggers generated by (Shen et al. 2021).
>
> **Q5**: There is no discussion and evaluation on adaptive attacks.
>
> **A**: We agree that one can develop attack methods targeting our approach. And meanwhile, we can further improve our method to address the adapted attacks. But we consider these developments beyond the scope of this paper. We will add some discussion along this direction in the final version of the paper.
>
> **Q6**: The paper does not clearly separate the contribution of existing works and this paper. For instance, the trigger reverse engineering technique by using a mask and a pattern in Section 3.1 is proposed by NC. Utilizing continuous values instead of binary values for the mask is also proposed by NC.
>
> **A**: We did not declare the reverse engineering pipeline as our contribution. What we proposed are the losses to improve the reverse engineering techniques. We made this more explicit in the revision. In Sec. 3.1, we clarified that ‘Our method is built on the existing reverse engineering pipeline first proposed by Neural Cleanse (Wang et al., 2019).’
>
> **Q7**: From the ablation results in Table 3, without the topological loss or the diversity loss, the results are already better than evaluated baselines. Do the authors care to explain in detail which component of the approach leads to the good results?
>
> **A**: The reason why the ablated version (our method without the new losses) is already better than neural cleanse is precisely because of the training with annotated models. We observe that even with only 25 annotated models, we can learn to maximally exploit the trigger information, and gain 10% AUC improvement. For more detailed discussion on supervised vs unsupervised, please refer to the General response Q1.
>
> (*Presentation issues*)
>
> **Q8**: The caption of Figure 5 is hard to be distinguished from the main text. There are no images with the injected triggers in Figure 6.
>
> **A**: Thanks for your suggestions. We have updated the captions of figure 5 (colored by blue). For the trojanAI task, we don’t have the ground truth -- the images with the injected triggers. Actually, figure 6 is an empirical illustration of the qualities of the recovered triggers by different methods. We will fix other presentation issues accordingly.

---

> > ### Comment · Reviewer_Tmo4 · 2021-11-29
> > **Thanks for the Response**
> >
> > Thanks for responding to my comments.
> >
> > Regarding applying the approach to other tasks, I was referring to training on one type of image classification task (e.g., CIFAR) and applying on another type (e.g., GTSRB). The current results presented in the submission use the same (train and test on CIFAR models) or similar (train and test on TrojAI models) tasks. However, in the real world, it is hard to generate a large set of models from the same or similar task for constructing a detection classifier. A small number of available models for training (e.g., 50) has a significant performance drop (from 0.92 to 0.81).
> >
> > It is important to show the resilience of any proposed defense techniques against possible potential attackers. If the defense method is only able to defend against known attacks, then one can easily develop a slightly modified attack to bypass any claimed security guarantee. There is no need to show the proposed defense is perfectly resilient to adaptive attacks but rather present the trend of the defense performance with the increase of attack strength.
> >
> > According to the response, the main performance gain is from training on a large set of annotated clean/poisoned models. This comes back to my previous concern regarding the practicality of such a defense mechanism. I would suggest to conduct some experiments to demonstrate the possibility of using models from different tasks for the training. For instance, train the detection classifier on CIFAR models and evaluate on GTSRB models. This will further strengthen the motivation.
> >
> > Minor issue: each poisoned TrojAI model comes with a set of clean and poisoned example images. Please include such examples images in Figure 6. Also, there is source code for generating the original TrojAI models and datasets. The authors can generate images in Figure 6 with the corresponding injected trigger. Please see https://github.com/trojai/trojai.
> >
> > With the above concerns unaddressed, I will keep my score.

---

> > > ### Author Response · Authors · 2021-11-30
> > > **Response to further feedback**
> > >
> > > Thanks very much for your feedback! Your suggestions will definitely strengthen our manuscript.
> > >
> > > We’d like to clarify that the main contribution of this paper is to use topological loss and diversity loss to improve the quality of the recovered triggers, in spite of supervised/unsupervised settings or number of training samples. The better recovered triggers contribute to better trojan detection performance, which has been validated in the table of General response (unsupervised setting), Table.2 (supervised setting) and Table.3 in the main text (supervised with different number of training models).
> > >
> > > > *The main performance gain is from training on a large set of annotated clean/poisoned models. A small number of available models for training (e.g., 50) has a significant performance drop (from 0.92 to 0.81).*
> > >
> > > First, the main point is to focus on the gain due to losses, not the gain due to training. The numbers are indeed supporting our claim: the proposed losses improve the detection performance despite the setting. In Table 3, we observe that with a small set of training models, the performance gain due to either topological loss or diversity loss is still up to 0.08 AUC (from 0.73 to 0.81 with 50 training models). We will add the baseline with both losses removed. We expect an even bigger gain compared to such a baseline in all settings.
> > >
> > > Second, although not quite the main point, we still consider 0.81 AUC significant, compared with the 0.63 AUC in the unsupervised setting. With only 50 training models, we already gain 0.81-0.63=0.18 AUC. This actually demonstrates the effectiveness of the proposed method and training strategy.
> > >
> > > Thanks for your suggestion regarding resilience. We agree that training on one type of image classification task and applying to another can show the resilience of a defense method. We will include such a study in the final version of the paper.
> > >
> > > We will update Fig.6 in the final version as you suggested.

---

### Official Review · Reviewer_3UGX · 2021-11-02

**Correctness:** 3
**Technical Novelty And Significance:** 4
**Empirical Novelty And Significance:** 3
**Recommendation:** 8
**Confidence:** 3

**Main Review:**

This paper proposed a novel target-label-agnostic reverse engineering method for reverse engineering in recovering trojaned triggers on clean images. The proposed diversity loss and topological prior significantly improved the performance in finding trojaned triggers and the quality of the found triggers compared to other existing methods. The proposed method is novel, the experiments were well conducted, and analysis was well performed.

**Summary Of The Paper:**

This paper proposes a diversity loss and a topological prior to not only increase the chances of finding the appropriate triggers but also improve the quality of the found triggers. These loss terms significantly improve the efficiency in finding trojaned triggers. The experiments results show that the proposed method performs substantially better than the baselines on the Trojaned-MNIST/CIFAR10 and TrojAI
datasets, respectively.

**Summary Of The Review:**

This paper proposed a novel target-label-agnostic reverse engineering method for reverse engineering in recovering trojaned triggers on clean images. he proposed diversity loss and topological prior significantly improved the performance in finding trojaned triggers and the quality of the found triggers compared to other existing methods. The proposed method is novel, the experiments were well conducted, and analysis was well performed.

---

> ### Author Response · Authors · 2021-11-23
> **Thanks for your positive feedback**
>
> **Q**: The proposed method is novel, the experiments were well conducted, and analysis was well performed.
>
> **A**: Thanks for your positive feedback!

---

### Public Comment · ~Yuri_Smirnov1 · 2021-11-09
**Missing citation**

A comment on missing citation. Since you are using the persistence diagrams of complexes, here is the reference for the paper where they were first introduced, under the name of canonical forms : Barannikov, S.(1994) "The Framed Morse Complex and its Invariants", Advances in Soviet Mathematics, 21: 93–115. Also, the algorithm for the computation of persistence diagrams, called "the classic algorithm" in your article, is described in section 2.1 in this paper.

---

### Author Response · Authors · 2021-11-23
**General response**

We thank all reviewers for their valuable feedback! We are glad that all reviewers appreciated the novelty of the contribution and the strong performance on challenging benchmarks. Below we address some common concerns.

**Q1**: 'Supervised setting does not seem practical./ Depending on model annotations is a limitation.' (Reviewer Tmo4, momo)

**A**: We would like to point out that our technical contributions are agnostic of whether the detection is supervised (i.e., using annotated models) or unsupervised. The proposed diversity and topological losses are used to improve the quality of a reconstructed trigger, using only a single model and a single input data. To show that the improved trigger quality leads to better Trojan detection, we evaluated our method on the supervised setting in the original manuscript.

Indeed, our method outperforms existing methods in different settings: fully supervised setting with annotated models, and unsupervised setting. In the original manuscript (Table 3), we have already demonstrated that with a limited number of annotated models, our method outperformed others. Below we report additional results in an unsupervised setting. To make a fair comparison, following Neural Cleanse and DLTND, we also use the simple technique based on Median Absolute Deviation (MAD) for trojan detection. We report the performance of Round 4 data of TrojAI here:

| Method | AUC | ACC |
|:------:|:------:|:------:|
|   NC  |  0.58  |    0.60   |
|  ABS  |  0.53 |   0.51   |
|    TABOR  |    0.52 |     0.55   |
|    ULP  |    0.54 |     0.57   |
|    DLTND  |    0.56 |     0.59   |
|    Ours  |    **0.63** |     **0.65**   |

Because of the better trigger quality, due to the proposed losses, our method outperforms baselines such as Neural Cleanse.

We also note that, in this table, all methods (including ours) perform unsatisfactorily in the unsupervised setting. This brings us back to the discussion as to whether a supervised setting is justified in Trojan detection (*although this is not directly relevant to our method*).

(**Unsupervised vs supervised**) From the research point of view, we believe that data-driven methods for Trojan detection are unavoidable as the attack techniques continue to develop. Like in many other security research problems, Trojan attack and defense are two sides of the same problem that are supposed to advance together. When the problem was first studied, classic unsupervised methods such as Neural Cleanse are sufficient. In recent years, the attack techniques have continued to develop, exploiting the entire dataset and leveraging techniques like adversarial training. Meanwhile, detection methods are confined with only the given model and a few sample data. For defense methods to move forward and to catch up with the attack methods, it seems only natural and necessary to exploit supervised approaches, e.g., learning patterns from public datasets such as the TrojAI benchmarks.

**Q2**: Compactness assumption: Are all the possible triggers compact? In other words, how does the introduced topological loss behave in case the adversary poisons the model with a scattered trigger?  (Reviewer 6VNg, momo)

**A**: This is a very good question. In TrojAI benchmarks, the assumption that the triggers are compact is true, as the triggers are polygons though varying in size and shape.

We'd like to discuss the adaptation of topological loss in a more general case. In theory, topological loss can be extended to a soft constraint and enforce a set of salient connected components to be recovered. This does not require a hard constraint on how many components a trigger consists of. In particular, recall the loss is based on the persistence diagram (Fig. 5 in the paper), in which each dot corresponds to one connected component. The saliency of a dot/component is measured by its persistence (i.e., difference between birth and death times). Theory has shown that high-persistence/salient components are more likely to come from the true structural signal, rather than noise (Edelsbrunner & Harer 2010, Cohen-Steiner et al. 2007). We currently choose to reconstruct the most salient one and to remove the rest. Alternatively, we can find a threshold of the persistence, reconstruct all dots/components with bigger persistence, and remove the rest.

This approach can be a future version of our method with a much more relaxed assumption of the trigger. We will allow an arbitrary number of components. The trigger can contain multiple scattered components. These components can be spatially far away from each other. This of course needs to be validated on a benchmark with scattered triggers.

Below we address specific reviews one-by-one.

---

### Decision · Program_Chairs · 2022-01-20

**Decision:**

Accept (Poster)

**Comment:**

This paper proposes a diversity loss and a topological prior to not only increase the chances of finding the appropriate triggers but also improve the quality of the found triggers. These loss terms significantly improve the efficiency in finding trojaned triggers. The experiments results show that the proposed method performs substantially better than the baselines on the Trojaned-MNIST/CIFAR10 and TrojAI datasets, respectively. This paper shows detailed ablation study with great empirical performance. Some reviewers have doubts about the experimental comparisons and some of the assumptions made in the algorithm. Overall, the thorough experimental investigation fo the proposed method makes this paper worthy of publication and widely being shared.